# Chemical range recognized by the ligand-binding domain in a representative amino acid-sensing taste receptor, T1r2a/T1r3, from medaka fish

Hikaru Ishida, Norihisa Yasui📗, Atsuko Yamashita📗*

Graduate School of Medicine, Dentistry and Pharmaceutical Sciences, Okayama University, Okayama, Japan

* a_yama@okayama-u.ac.jp

**Data Availability Statement:** All relevant data are within the manuscript and its Supporting Information files.

## Abstract

Taste receptor type 1 (T1r) proteins are responsible for recognizing nutrient chemicals in foods. In humans, T1r2/T1r3 and T1r1/T1r3 heterodimers serve as the sweet and umami receptors that recognize sugars or amino acids and nucleotides, respectively. T1rs are conserved among vertebrates, and T1r2a/T1r3 from medaka fish is currently the only member for which the structure of the ligand-binding domain (LBD) has been solved. T1r2a/T1r3 is an amino acid receptor that recognizes various L-amino acids in its LBD as observed with other T1rs exhibiting broad substrate specificities. Nevertheless, the range of chemicals that are recognized by T1r2a/T1r3LBD has not been extensively explored. In the present study, the binding of various chemicals to medaka T1r2a/T1r3LBD was analyzed. A binding assay for amino acid derivatives verified the specificity of this protein to L-α-amino acids and the importance of α-amino and carboxy groups for receptor recognition. The results further indicated the significance of the α-hydrogen for recognition as replacing it with a methyl group resulted in a substantially decreased affinity. The binding ability to the protein was not limited to proteinogenic amino acids, but also to non-proteinogenic amino acids, such as metabolic intermediates. Besides L-α-amino acids, no other chemicals showed significant binding to the protein. These results indicate that all of the common structural groups of α-amino acids and their geometry in the L-configuration are recognized by the protein, whereas a wide variety of α-substituents can be accommodated in the ligand binding sites of the LBDs.

## Introduction

Animals sense a wide variety of chemicals in foods via taste receptors in their oral cavity. Numerous taste substances can be categorized into five basic taste modalities: sweet, umami, bitter, salty, and sour. The chemicals in each modality are recognized by the specific taste receptors [1]. For example, nutrient substances, such as sugars, amino acids, and nucleotides, are recognized by taste receptor type 1 (T1r) proteins. In humans and mice, the T1r1/T1r3 heterodimer serves as an umami receptor responsible for detecting amino acids and nucleotides,

**Funding:** This work was financially supported by JSPS KAKENHI Grant Numbers JP20H03195, JP20H04778, JP21H05524, and JP23H02424, and the Takeda Science Foundation (to A.Y.). The funders had no role in study design, data collection and analysis, decision to publish, or preparation of the manuscript.

**Competing interests:** The authors have declared that no competing interests exist.

whereas the T1r2/T1r3 heterodimer serves as a sweet receptor responsible for detecting sugars [2–4]. The ligand specificities of T1r proteins are not strictly conserved among animals and there are some varieties. For example, all T1rs in zebrafish and medaka fish recognize amino acids but not sugars [5].

T1rs belong to the class C G protein-coupled receptor (GPCR) family, which is characterized by the presence of a large ligand-binding domain (LBD) in the extracellular region [6, 7]. The LBD contains a binding site for intrinsic receptor agonists, known as the orthosteric ligand-binding site, in the middle cavity of the domain with a bi-lobal architecture. The intrinsic agonists of T1rs (i.e., sugars, amino acids, and nucleotides) are also recognized at the orthosteric sites in T1rs [8–12]. Nevertheless, the orthosteric sites in T1rs have distinctive properties compared with those in other class C GPCRs. The former is amenable to accommodate a certain range of chemicals, whereas the latter shows specificity for limited compounds. For example, the human sweet receptor T1r2/T1r3 responds to sucrose as well as artificial sweeteners, such as aspartame, saccharin, and acesulfame K, all of which are recognized by the orthosteric binding site in T1r2-LBD [9, 10]. The mouse T1r1/T1r3 also recognizes a wide array of L-amino acids at the orthosteric binding site of T1r1LBD [11]. These properties are in contrast with those of other class C GPCRs responsible for recognizing signaling molecules, such as metabotropic glutamate receptors, which exhibit strict specificity for L-glutamate [13]. Nevertheless, the broad substrate specificities associated with T1rs correspond to their physiological functions in recognizing various chemicals in the environment using a limited set of receptors. However, these characteristics raise a question of how broadly a range of chemicals can be recognized by the orthosteric site of a particular T1r.

Recent studies have indicated that T1rs are expressed, not only in the oral cavity, but also in many other organs in the body, such as the brain, gastrointestinal system, and reproductive organs [14, 15]. Therefore, these receptors are likely responsible for various physiological functions in these tissues. However, only a fraction of them has been elucidated thus far. Determining the range of substrates for T1rs will provide clues for understanding the physiological functions of T1rs in the body, as well as in taste sensation.

In this study, we examined ligands for T1r2a/T1r3LBD from medaka (*Oryizias latipes*) as a case study for determining the substrate specificity of a T1r. Medaka T1r2a/T1r3 is an amino acid receptor and the sole T1r protein for which ligand binding and structural analyses using a heterodimeric recombinant protein have been achieved [12, 16, 17]. Medaka T1r2a/T1r3LBD reportedly accommodates a wide range of L-amino acids in orthosteric binding sites [5, 12, 17] and is considered a suitable target protein to address the broad substrate specificities of T1rs. To assess the ligand-binding activity of this protein, we used differential scanning fluorimetry (DSF). In this method, binding is evaluated by the extent of ligand-dependent thermal stabilization of the target protein, which is judged by a high-temperature shift of the melting curves of thermal denaturation detected using an environmentally sensitive fluorescent dye as an indicator [18]. We previously verified that the results of ligand-binding to T1r2a/T1r3LBD analyzed using DSF were consistent with those obtained using other biophysical methodologies [17]. In this study, we performed an assay on an extended range of chemicals using this method and clarified the specificity of this protein.

## Materials and methods

Protein samples were prepared as previously described [12, 16]. Briefly, the heterodimer of the C-terminal FLAG-tagged T1r2aLBD and T1r3LBD [12, 19] was stably expressed in *Drosophila* S2 cells (Invitrogen) and purified from the culture medium using ANTI-FLAG M2 affinity gel (SIGMA).

The ligand binding assay by differential scanning fluorimetry (DSF) and data analysis were performed as described previously [17]. Briefly, the purified protein was dialyzed against the assay buffer (20 mM Tris-HCl, 300 mM NaCl, 2 mM CaCl$_2$, pH 8.0). The sample protein was then mixed with Protein Thermal Shift Dye (Applied Biosystems) and the ligands in the dialysis buffer used for the final dialysis step. Specifically, 1 μg of protein, 1× Dye, and ligands at a concentration for each condition were mixed in 20 μL of reaction mix. Prior to mixing, the ligands, except for steroids, were dissolved in the same solution and the pH was adjusted to 8.0 through the addition of either NaOH or HCl. Because of the low solubilities in aqueous solution, the steroids were first dissolved in dimethyl sulfoxide (DMSO) and added to the reaction mixture at a final DMSO concentration of 10%. Fluorescence intensity changes accompanying the thermal denaturation of the protein were measured using the StepOne Real-Time PCR System (Applied Biosystems) with heating of the sample at the temperature from 25°C to 99°C at a rate of 0.022°C/s. The apparent melting temperature ($T_m$) of each sample was determined using the maximum of the derivatives of the melting curve (dFluorescence/dT). To ensure sample activity, data from the sample that did not meet the criteria that the $T_m$ in the absence of a ligand was within 51–55°C and the $T_m$ increase ($\Delta T_m$) in the presence of 10 mM L-alanine was within 9–11°C were excluded.

The apparent dissociation constant ($K_{d\text{-app}}$) of the ligand was estimated by fitting the $T_m$ values at different ligand concentrations to an equation based on the thermodynamic folding model proposed by Schellman [20] using KaleidaGraph (Synergy Software) as described previously [17].

## Results

### Binding of amino acids and their derivatives

We previously reported that medaka T1r2a/T1r3LBD binds to a variety of L-amino acids [12, 17]. To further address amino acid specificity, the binding of alanine and its derivatives (Fig 1A) to T1r2a/T1r3LBD was examined.

As observed previously [17], T1r2a/T1r3LBD showed thermal stabilization, as evidenced by an increase in $T_m$ ($\Delta T_m$), which was dependent on the L-alanine concentration, whereas no significant $\Delta T_m$ change was observed with the addition of D-alanine (Fig 1B and S1 Fig). These results indicated substantial binding of the former and low or negligible binding of the latter, confirming the L-amino acid specificity of this receptor. Next, we addressed the requirement of each amino acid functional group by testing chemicals devoid of the α-carboxy or amino group of alanine, namely ethylamine and propionate. These results indicated that they did not induce thermal stabilization of the protein, indicating no significant binding. Therefore, both the α-carboxy and the amino group in L-amino acids are essential for recognition by T1r2a/T1r3LBD.

We also addressed the recognition stringency of each amino acid functional group. The L-alanine derivatives, in which the α-carboxy and the amino group are modified [i.e., *N*-methyl-L-alanine (*N*-Me-L-Ala) and L-alanine methyl ester (L-Ala Me ester; Fig 1A)], still showed substantial binding signals, whereas their D-enantiomers exhibited little signal (Fig 1B). In addition, 2-methylalanine (2-Me Ala), an achiral alanine derivative in which the hydrogen at the Cα is substituted with a methyl group (Fig 1A), exhibited weaker binding signals. The apparent $K_d$ ($K_{d\text{-app}}$) of these derivatives derived from a dose-thermal stabilization analysis (Fig 1C) revealed that modification at the α-carboxy or amino group (*N*-Me-L-Ala and L-Ala Me ester) increased the $K_{d\text{-app}}$ by approximately 10-fold ($\sim$600–1100 μM) compared to the non-modified L-alanine ($K_{d\text{-app}}$: $\sim$60 μM), whereas modification at the Cα (2-Me Ala) resulted in a >100-fold increase ($K_{d\text{-app}}$: > 10 mM; Fig 1C, Table 1). These results suggest that the α-

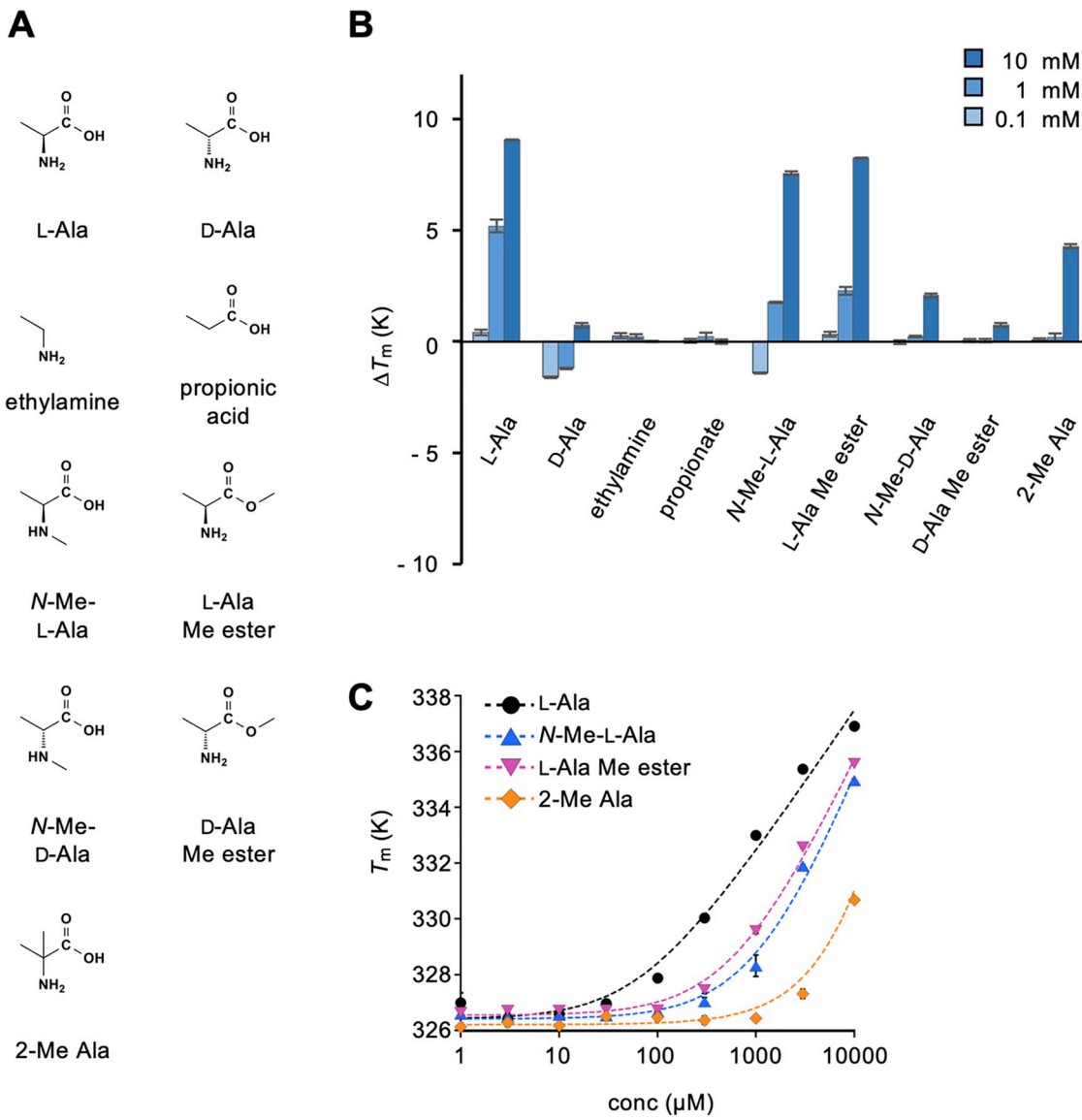

**Fig 1. Binding analysis of amino acids and their derivatives to T1r2a/T1r3LBD using differential scanning fluorimetry.** (A) Chemical structures of amino acids and their derivatives tested. (B) $\Delta T_m$ resulting from the addition of 0.1, 1, and 10 mM amino acids and their derivatives. (C) Dose-dependent $T_m$ changes caused by the addition of L-alanine and its derivatives. Data represent the mean ± standard error of the mean (SEM) of four technical replicates. The fitted line drawn for the 2-methylalanine data assumed the $K_{d\text{-app}}$ as 15 mM, although accurate $K_{d\text{-app}}$ determination was impractical owing to a large fitting error.

**Table 1. Apparent dissociation constants ($K_{d\text{-app}}$) of L-alanine derivatives to medaka T1r2a/T1r3LBD as determined by differential scanning fluorimetry.**

| ligands | $K_{d\text{-app}}$ (μM) |
|---|---|
| L-alanine | 63.9 ± 29.0 |
| *N*-methyl-L-alanine | 1140 ± 407 |
| L-alanine methyl ester | 615 ± 154 |
| 2-methylalanine | ND |

carboxy and the amino groups of amino acids are important for recognition by T1r2a/T1r3LBD, and that their modification impairs recognition to some extent, but not critically. Nevertheless, the modification at Cα results in severe consequences for binding compared with those at the α-carboxy and the amino groups, indicating the significance of the α-carbon and/or hydrogen for T1r2a/T1r3LBD recognition.

## Binding of ligands for the other T1rs or class C GPCRs

We examined the commonality in ligand specificities among T1rs and other class C GPCRs. A recent study reported that fish T1r2 and T1r3 are not orthologs but paralogs of mammalian Tlr receptors [21]. We examined the binding of sugars (Fig 2A), which are substrates of the mammalian sweet receptors T1r2/T1r3 [2, 4], to medaka T1r2a/T1r3LBD. Glucose, fructose, and sucrose as well as an artificial sweetener sucralose, as a sugar analog, did not exhibit significant binding (Fig 2B). It should be noted that it is not feasible to assess the specific binding of sugars at high concentrations, such as those inducing responses of mammalian T1r2/T1r3 (~300 mM) [2, 4], using DSF, because they intrinsically induce thermal stabilization of a protein, even without specific interactions [22]. Nevertheless, the observation in this study, which showed no noticeable thermal stabilization by sugars up to at least 10 mM, is consistent with the results of a previous study that reported no responses of this receptor to sugars (at 150 mM) [5]. Next, we examined the binding of a nucleotide, inosine monophosphate (IMP), because the mammalian amino acid receptors T1r1/T1r3 recognize IMP as a substrate and exhibit enhanced responses to amino acids in its presence [3, 8]. However, in medaka T1r2a/T1r3LBD, IMP did not bind, and its presence did not enhance alanine binding (Fig 2B).

One of the class C GPCRs, calcium sensing receptor (CaSR) responds to polyamines and aminoglycoside antibiotics [23] (Fig 3A). We examined the binding of putrescine, spermidine, and spermine as polyamines and gentamicin, neomycin, kanamycin, and spectinomycin as

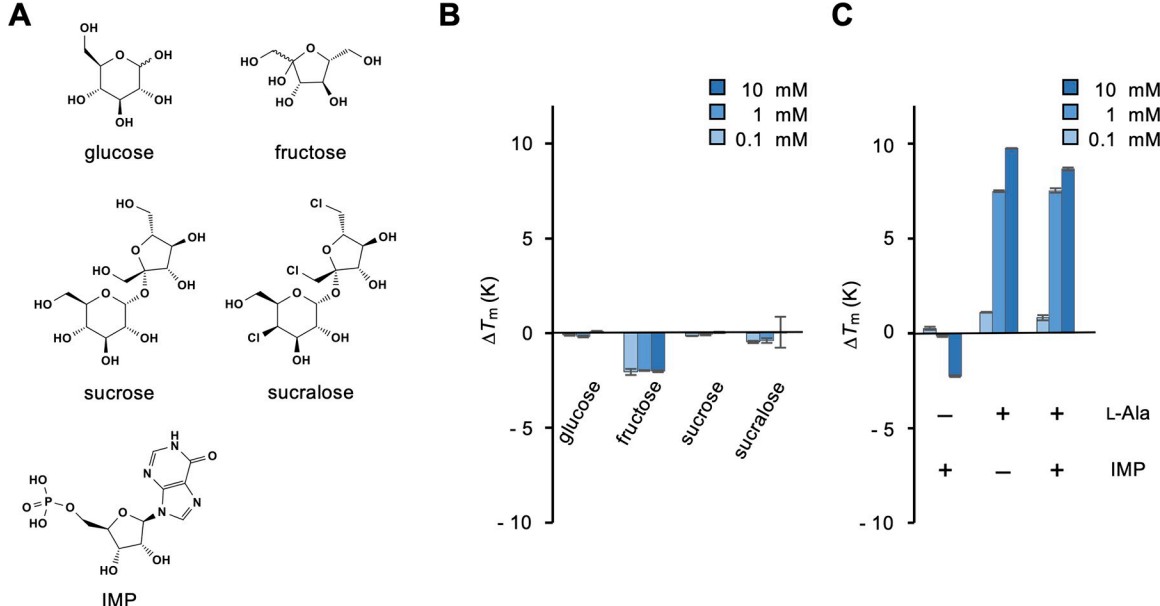

**Fig 2. Binding analysis of typical ligands for T1rs to T1r2a/T1r3LBD.** (A) Chemical structures of the T1r ligands tested. (B) $\Delta T_m$ resulting from the addition of 0.1, 1, and 10 mM sugars. (C) $\Delta T_m$ resulting from the addition of 0.1, 1, and 10 mM IMP (left), L-alanine (middle), and L-alanine in the presence of 1 mM IMP (right). Data represent the mean ± SEM of four technical replicates.

aminoglycoside antibiotics to medaka T1r2a/T1r3LBD; however, none of these compounds exhibited binding signals (Fig 3B).

In summary, we could not find any ligands of medaka T1r2a/T1r3 common to the other T1rs and class C GPCRs other than L-amino acids.

## Binding of bioactive compounds

To explore the physiological functions of T1rs in the body, we examined the binding of various bioactive compounds that may serve as substrates in organs other than the oral cavity.

We tested several amino acid-derived metabolites, including L-ornithine, L-citrulline, L-carnitine, and taurine (Fig 4A). L-citrulline and, albeit weakly, L-ornithine, which are metabolites of the urea cycle, showed obvious binding to medaka T1r2a/T1r3LBD (Fig 4B). The results indicated that α-amino acid metabolites serve as ligands for T1r2a/T1r3LBD, even if they are non-proteinogenic amino acids. L-carnitine and taurine, which are biosynthesized from L-amino acids but do not have α-amino acid structures, exhibited no obvious binding to the protein (Fig 4B), again indicating the significance of the α-amino acid structure required for recognition. Other non-amino acid metabolites or biosynthetic precursors, such as glucosamine and citrate, did not induce thermal stabilization of T1r2a/T1r3LBD (Fig 4A and 4B). Other nutrient chemicals were tested, including several water-soluble vitamins, such as ascorbate, thiamine, and nicotinate (Fig 5A); however, none exhibited binding to medaka T1r2a/T1r3LBD (Fig 5B).

Finally, we tested small signaling molecules, such as neurotransmitters and hormones, considering the fact that T1rs are expressed in the brain and reproductive organs [24, 25] (Fig 6A). Thermal stabilization of T1r2a/T1r3LBD was not observed following the addition of representative neurotransmitters, such as serotonin as a representative of monoamines, γ-aminobutyric acid (GABA), and adenosine triphosphate (ATP) (Fig 6B). The finding that GABA was not recognized by T1r2a/T1r3LBD confirms the specificity of the protein to α-amino acids, but not γ-amino acids. We also examined the binding of representative steroids, testosterone and

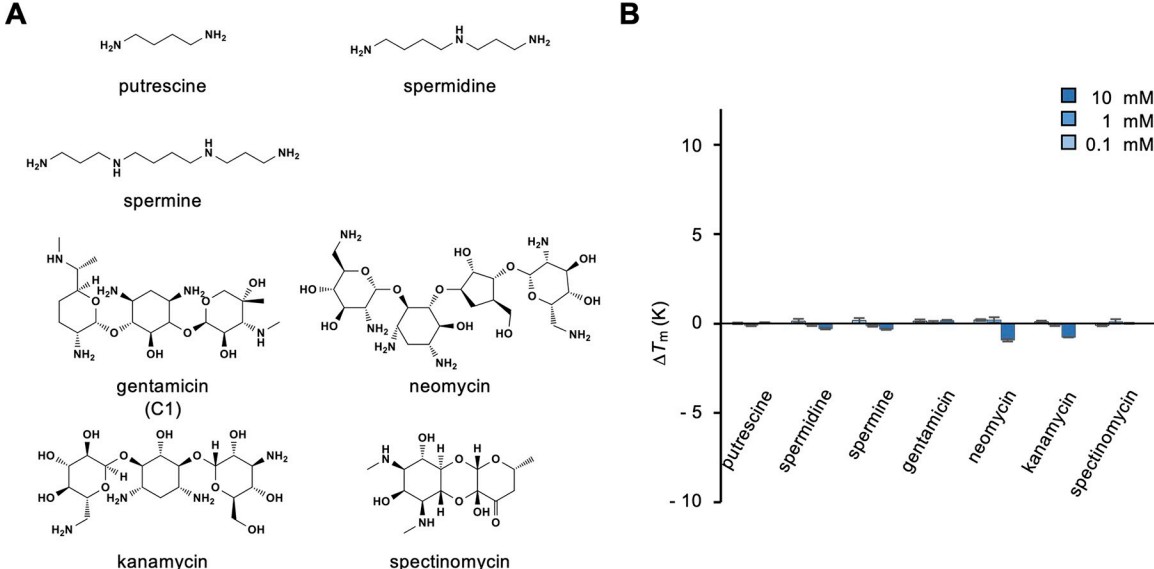

**Fig 3. Binding analysis of typical CaSR ligands to T1r2a/T1r3LBD.** (A) Chemical structures of the CaSR ligands tested. The structure of the C1 component of the gentamicin complex is shown in the panel, whereas the complex reagent was used for the analysis. (B) $\Delta T_m$ resulting from the addition of 0.1, 1, and 10 mM CaSR ligands. Data represent the mean ± SEM of four technical replicates.

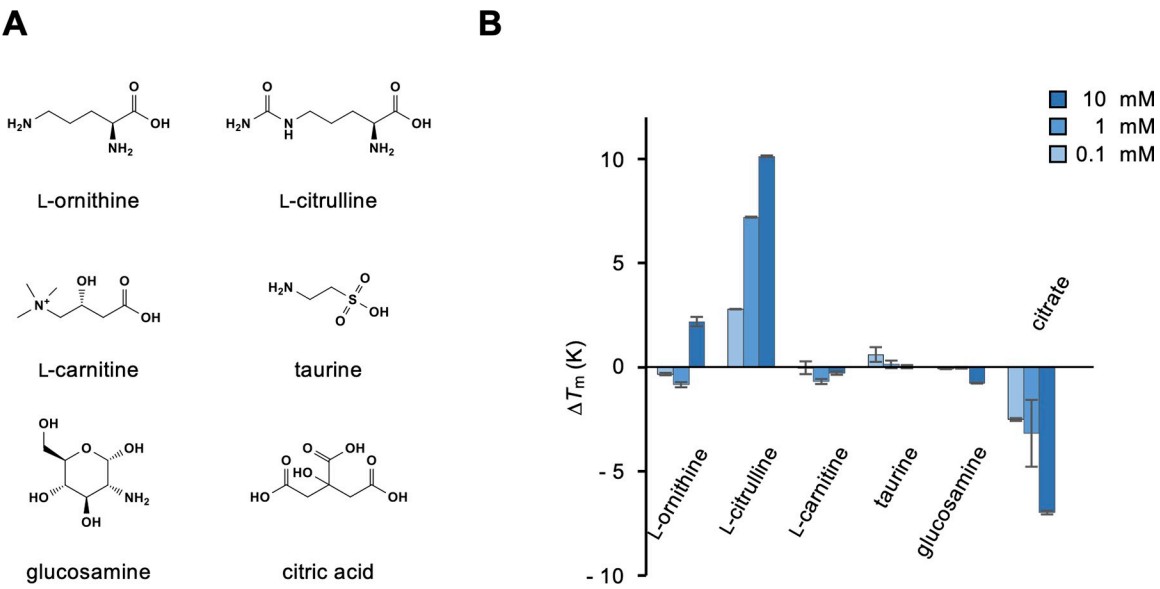

**Fig 4. Binding analysis of representative amino acid and non-amino acid metabolites and biosynthetic precursors to T1r2a/T1r3LBD.** (A) Chemical structures of the metabolites and biosynthetic precursors tested. (B) $\Delta T_m$ resulting from the addition of 0.1, 1, and 10 mM metabolites or biosynthetic precursors. Data represent the mean ± SEM of four technical replicates.

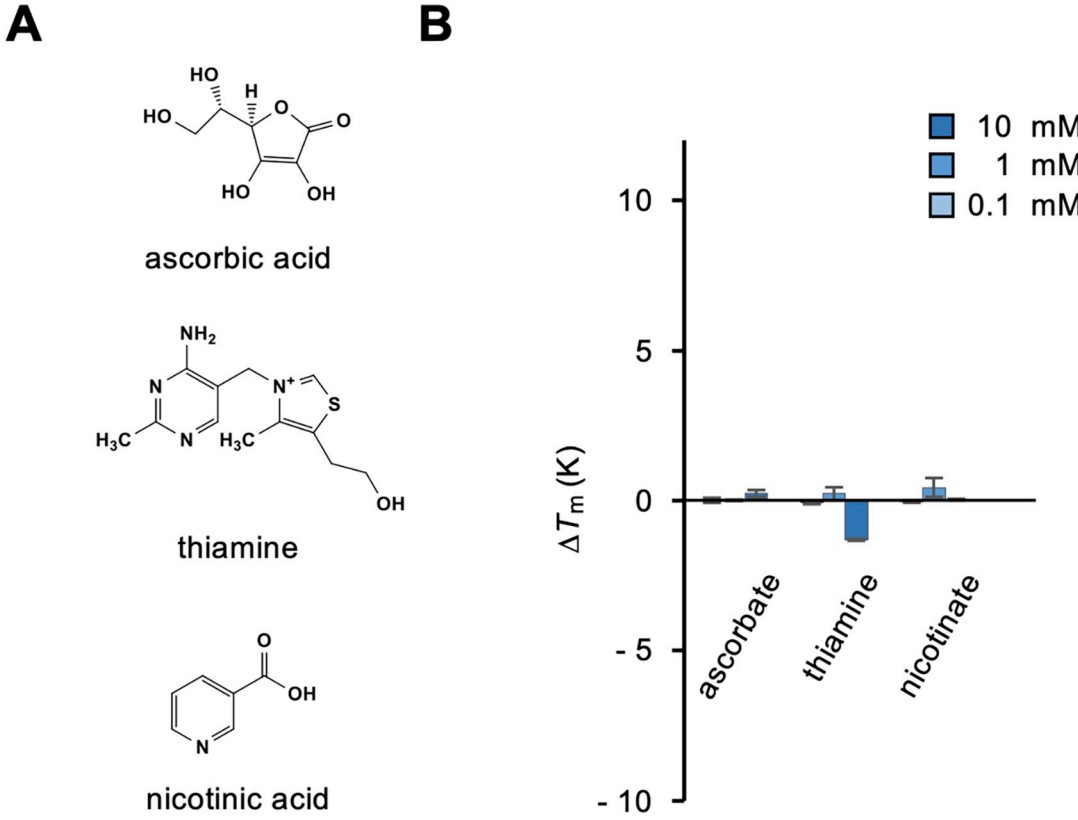

**Fig 5. Binding analysis of representative water-soluble vitamins to T1r2a/T1r3LBD.** (A) Chemical structures of the vitamins tested. (B) $\Delta T_m$ resulting from the addition of 0.1, 1, and 10 mM vitamins. Data represent the mean ± SEM of four technical replicates.

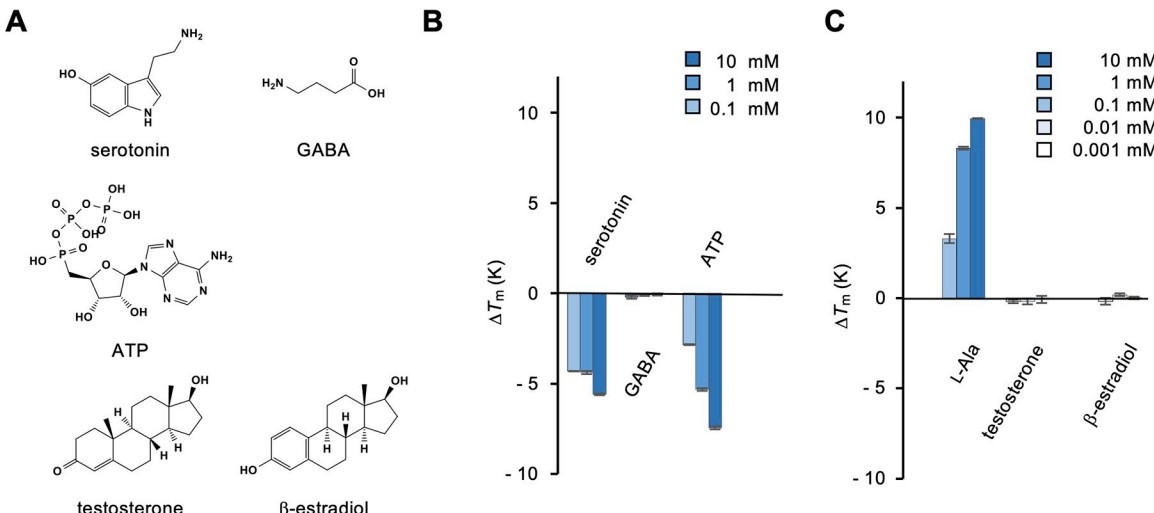

**Fig 6. Binding analysis of representative neurotransmitters and hormones to T1r2a/T1r3LBD.** (A) Chemical structures of the neurotransmitters and hormones tested. (B) $\Delta T_m$ resulting from the addition of 0.1, 1, and 10 mM neurotransmitters. (C) $\Delta T_m$ resulting from the addition of 0.001, 0.01, and 0.1 mM hormones in 10% DMSO. $\Delta T_m$ values resulting from the addition of 0.1, 1, and 10 mM L-alanine in 10% DMSO are also shown for comparison. Data represent the mean ± SEM of four technical replicates.

β-estradiol (Fig 6A). Because steroids exhibited low solubility in the assay buffer, before testing the binding of the steroids in a 10% DMSO buffer, we first confirmed that T1r2a/T1r3LBD showed similar thermal stabilization with the addition of L-alanine in the presence of 10% DMSO to that observed without DMSO (Fig 6C compared with Fig 1B; S1 Fig). Despite testing concentrations up to 0.1 mM, which is the highest soluble concentration for these chemicals in 10% DMSO, no significant binding of these steroids was observed (Fig 6C).

Taken together, among the tested compounds, no bioactive compounds other than L-α-amino acids were identified as ligands for medaka T1r2a/T1r3LBD. Nevertheless, the results also indicate that the L-α-amino acid specificity of T1r2a/T1r3LBD is not limited to the proteinogenic amino acids examined thus far but also to non-proteinogenic amino acids, such as metabolic intermediates and products.

## Discussion

In this study, we examined the ligand specificity of medaka T1r2a/T1r3LBD. The binding assay confirmed the specificity of L-α-amino acid binding to this protein, whereas no significant binding of other chemicals lacking the L-α-amino acid architecture was observed. More specifically, the results indicated the importance of α-amino and carboxy groups in amino acids for recognition by the receptor as observed in the crystallographic structure [12], although small modifications, such as methylation, were found to reduce the affinities but still resulted in binding (Fig 1). Furthermore, the results of the present study indicate the significance of the α-hydrogen for the recognition of amino acids, as replacing this hydrogen with a methyl group resulted in decreased affinity (Fig 1). Because the α-hydrogen in the bound amino acid points toward the aromatic residues just below the substrate, Phe213 in T1r2a and Tyr221 in T1r3 (Fig 7A and 7B), the results indicate that not only the hydrophobic interaction between the ligand and the aromatic residue but also the CH-π interaction between the α-hydrogen of the ligand and the side-chain aromatic ring in the protein is significant for recognition. The manner of recognition explains the L-α-amino acid specificity of the receptor: all of the common structural components of α-amino acids, α-hydrogen, amino, and carboxy

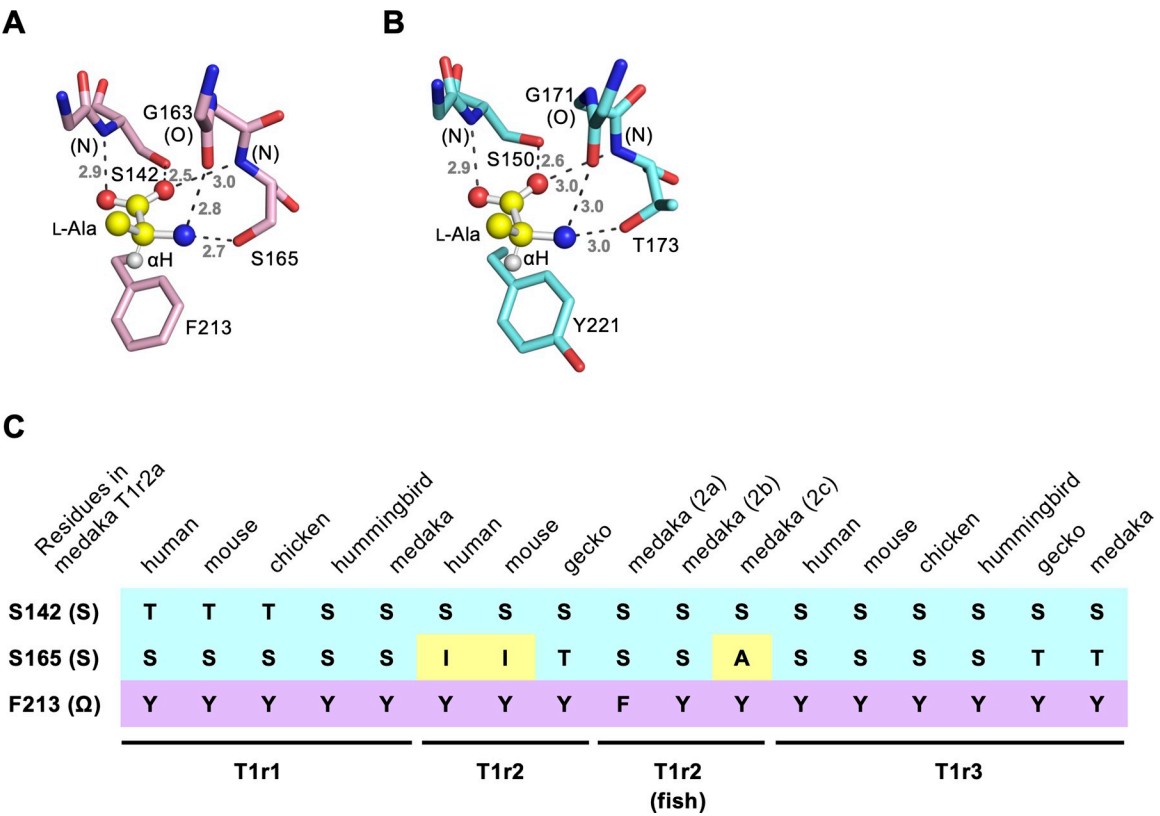

**Fig 7. Residues important for the interaction with L-amino acids at the ligand-binding sites in T1r2a/T1r3LBD.** (A, B) Close-up views of the ligand-binding sites in T1r2a (A) and T1r3 (B) in the crystal structure of L-alanine-bound T1r2a/T1r3LBD (PDB ID: 5X2N) [12]. Bound L-alanine is shown as a ball-and-stick model. The dashed lines depict the hydrogen bonds labeled with their distances in Å. (C) The "SSΩ" residues corresponding to Ser142, Ser165, and Phe213 in T1r2a for various T1rs. Representative T1rs, in which responses have been reported, are shown [2–5, 11, 12, 26, 27].

groups, and their geometry in the L-configuration, are recognized by the protein. If these groups are present, medaka T1r2a/T1r3 can accommodate a variety of α-substituents in its large ligand-binding pocket [12]. Indeed, we found that the binding ability was not limited to proteinogenic amino acids [17] but also to non-proteinogenic amino acids, such as metabolic intermediates and products (Fig 4). Docking simulations showed that these compounds could be accommodated in the ligand-binding site in a manner similar to the that of known ligands (e.g., L-alanine) without steric clashes (S2 Fig).

The manner of amino acid recognition in medaka T1r2a/T1r3 is likely conserved among T1rs that share amino acid-responding functions (Fig 7C). In medaka T1r2a, the side-chain hydroxyl groups of Ser142 and Ser165, together with the main-chain amide and carbonyl groups as well as that in another residue (Gly163 in T1r2a), form a hydrogen-bonding network with the α-amino and carboxy groups in the ligand amino acid (Fig 7A). These two Ser/Thr (S) residues, as well as the aromatic residue (Ω; Phe213 in T1r2a) that forms a CH-π interaction with the α-hydrogen in the ligand, are conserved among most T1rs (Fig 7B and 7C). Therefore, the ability to recognize L-α-amino acids may be common to T1rs that share these three "SSΩ" residues at the orthosteric binding site. An exception without strict conservation of the "SSΩ" residues was observed for mammalian T1r2, which has an isoleucine at the residue corresponding to Ser165 [12]. This may be related to the fact that human and mouse sweet receptors T1r2/T1r3 respond to sugars and D-amino acids, but not to L-amino acids [2, 4].

Another exception is medaka T1r2c, which contains an alanine at the same residue. This may be related to the fact that medakaT1r2c/T1r3 specifically responds to L-proline, a secondary amino acid, but not to other L-amino acids with a primary amino group [5, 28]. Conversely, hummingbird T1r1 and gecko T1r2 share "SSΩ" residues, whereas hummingbird T1r1/T1r3 and gecko T1r2/T1r3 exhibit sugar responses [26, 27]. This suggests that the "SSΩ" residues are compatible with the sugar responding function. Nonetheless, the actual receptor functions need to be examined experimentally.

In contrast to the recognition of common groups in L-α-amino acids, the manner of recognition of the α-substituent group likely varies among T1rs, as evidenced by the different substrate specificities observed. For example, human T1r1/T1r3 specifically responds to acidic amino acids, such as L-glutamate and aspartate, whereas mouse T1r1/T1r3 responds to a wide array of L-amino acids except acidic amino acids [3, 4, 11]. Therefore, the range of amino acid specificity for T1r is likely determined by the physicochemical properties of the ligand-binding pocket surrounding the α-substituent group in each receptor. For T1rs sharing broad ligand specificity, such as mouse T1r1/T1r3 and medaka T1r2a/T1r3, the ligands may not be limited to proteinogenic amino acids, but also non-proteinogenic amino acids, as observed for medaka T1r2a/T1r3. This warrants further investigation by binding assays that involve a wider range of chemicals.

Besides L-α-amino acids, no other chemicals tested in this study exhibited significant binding to medaka T1r2a/T1r3LBD. Although the addition of several compounds such as IMP, citric acid, thiamine, serotonin, and ATP provided negative $\Delta T_m$ values, their actions on the protein are unlikely related to the binding that induces receptor responses, but probably destabilization of the protein due to their acidic/basic properties, because the correlation between the positive $\Delta T_m$ values and the receptor responses has been reported [17]. On this account, a representative monoamine neurotransmitter serotonin did not show binding expected to induce receptor responses (Fig 6B), which is consistent with the results that T1r2a/T1r3 prefers α-amino acids and not monoamines (ethylamine; Fig 1). Similarly, chemicals sharing chemical structures that are distinct from those of amino acids, such as steroids, did not show obvious binding. However, because the assay was performed solely using the LBD and not the full-length receptor because of methodological limitations, the possibility that these molecules act on other regions in the receptor cannot be excluded. On the other hand, the newly identified L-α-amino acid ligands, L-citrulline and L-ornithine, are cytosolic metabolic intermediates in the urea cycle, whereas they exhibit binding to the extracellular LBD of T1r2a/T1r3. Therefore, it is unclear whether they act as intrinsic ligands for the receptor in vivo. Moreover, the actions of these chemicals on T1r2a/T1r3 as agonists or antagonists are also unclear, although we verified that L-α-amino acid binding analyzed by DSF correlates with the receptor response induced by the ligand [17]. Nonetheless, our results indicate the possibility that a wider range of chemicals beyond those considered may be significant for gustation and could serve as ligands for T1rs. Their physiological relevance should be examined in future studies.

## Supporting information

**S1 Fig. Thermal melting curves of T1r2a/T1r3LBD.** (A) Representative melting curves in the assay buffer. The $T_m$ values derived from four technical replicated measurements represent no ligand, 53.0 ± 0.16˚C; 0.1 mM L-alanine added, 53.4 ± 0.12˚C ($\Delta T_m$, 0.4˚C); 1 mM L-alanine added, 58.2 ± 0.29˚C ($\Delta T_m$, 5.2˚C); 10 mM L-alanine added; 62.0 ± 0.02˚C ($\Delta T_m$, 9.0˚C). (B) Representative melting curves in the assay buffer containing 10% DMSO. The $T_m$ values derived from four technical replicated measurements represent no ligand, 47.3 ± 0.23˚C 0.1 mM L-alanine added, 50.9 ± 0.50˚C ($\Delta T_m$, 3.6˚C); 1 mM L-alanine added, 55.9 ± 0.16˚C ($\Delta T_m$,

8.6˚C); 10 mM L-alanine added; 57.5 ± 0.02˚C ($\Delta T_m$, 10.2˚C).
(TIF)

**S2 Fig. Docking simulations of the compounds revealed binding to T1r2a/3LBD in this study.** (A) A close-up view of the ligand-binding site in T1r2a in the crystal structure of the L-alanine-bound form (PDB ID: 5X2N). (B–D) Representative docking poses of *N*-methyl-L-alanine (*N*-Me-L-Ala; ZINC 901468; B), L-alanine methyl ester (L-Ala Me ester; ZINC 34702232; C), and L-citrulline (ZINC 1532614; D) in the ligand-binding site of T1r2a. In panels B–D, the docking simulations were performed using SwissDock (Grosdidier *et al. Nucleic Acids Res.* 39, W270, 2011), using the coordinates of T1r2aLBD (PDB ID: 5X2N, chain A) without ligands as a target.
(TIF)

**S1 File. The file archive including all the data reported in the main text and the Supporting information.** SourceData.xlsx, the data for reproducing the figures in the main text and S1 Fig; 5x2n_A_nolig.pdb, the coordinates of T1r2aLBD used as the input for docking simulations; LNMA.pdb, the coordinates for a representative docked model of *N*-methyl-L-alanine, shown in S2B Fig; LAME.pdb, the coordinates for a representative docked model of L-alanine methyl ester, shown in S2C Fig; Lcit.pdb, the coordinates for a representative docked model of L-citrulline, shown in S2D Fig.
(ZIP)

## Acknowledgments

We thank Junya Nitta, Rakuto Mizoguchi, and Mayu Kudo for their help with the protein preparation.

## Author Contributions

**Conceptualization:** Atsuko Yamashita.

**Formal analysis:** Hikaru Ishida, Atsuko Yamashita.

**Funding acquisition:** Atsuko Yamashita.

**Investigation:** Hikaru Ishida, Atsuko Yamashita.

**Methodology:** Norihisa Yasui.

**Supervision:** Atsuko Yamashita.

**Visualization:** Hikaru Ishida, Atsuko Yamashita.

**Writing – original draft:** Hikaru Ishida, Atsuko Yamashita.

**Writing – review & editing:** Hikaru Ishida, Norihisa Yasui, Atsuko Yamashita.

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
