## [Decision Letter · Decision Letter 0]

23 Jan 2024

PONE-D-23-32338Chemical range recognized by the ligand-binding domain in a representative amino acid-sensing taste receptor, T1r2a/T1r3, from medaka fishPLOS ONE

Dear Dr. Yamashita,

Thank you for submitting your manuscript to PLOS ONE. After careful consideration, we feel that it has merit but does not fully meet PLOS ONE’s publication criteria as it currently stands. Therefore, we invite you to submit a revised version of the manuscript that addresses the points raised during the review process.

Overall, the three reviewers assessed your manuscript favorably. But Reviewer 2 expressed some minor concerns that you are requested to address in full in your revised manuscript.

We look forward to receiving your revised manuscript.

Kind regards,

Israel Silman

Academic Editor

PLOS ONE

Journal Requirements:

4. Thank you for stating the following financial disclosure: "This work was financially supported by JSPS KAKENHI Grant Numbers JP20H03195, JP20H04778, JP21H05524, and JP23H02424 (to A.Y.)."  

5. Thank you for stating the following in the Acknowledgments Section of your manuscript: "This work was financially supported by JSPS KAKENHI Grant Numbers JP20H03195, JP20H04778, JP21H05524, and JP23H02424 (to A.Y.)." 

Please remove any funding-related text from the manuscript and let us know how you would like to update your Funding Statement. Currently, your Funding Statement reads as follows: "This work was financially supported by JSPS KAKENHI Grant Numbers JP20H03195, JP20H04778, JP21H05524, and JP23H02424 (to A.Y.)." 

Reviewers' comments:

Reviewer's Responses to Questions

**Comments to the Author**

1. Is the manuscript technically sound, and do the data support the conclusions?

Reviewer #1: Yes

Reviewer #2: Yes

Reviewer #3: Yes

2. Has the statistical analysis been performed appropriately and rigorously? 

Reviewer #1: Yes

Reviewer #2: N/A

Reviewer #3: Yes

3. Have the authors made all data underlying the findings in their manuscript fully available?

Reviewer #1: Yes

Reviewer #2: Yes

Reviewer #3: Yes

4. Is the manuscript presented in an intelligible fashion and written in standard English?

Reviewer #1: Yes

Reviewer #2: Yes

Reviewer #3: Yes

5. Review Comments to the Author

Reviewer #1: Ishida et al reports on binding assays aimed at determining the ligand binding specificity of the medaka fish T1r2a/T1r3 ligand binding domain (LBD). The crystallographic structure of this dimer has been solved, which enables more educated inferences from this study. The study is carefully conducted, and they use a variety of structurally diverse alanine derivatives, non-proteogenic amino acids, and structurally unrelated sugars, neurotransmitters, hormones, and vitamins. They conclude that only L amino acids are capable of binding to the LBD, and that substitutions to the carboxyl and amino groups are more tolerated compared to the substitution of the hydrogen at the alpha carbon. The paper is well written, and the conclusions are well supported by the experiments. My only comment is that they have used very low concentrations of glucose and sucrose. Although these are not natural ligands of fish T1rs, sucrose is known to generate some nerve response at a high concentration of around 300 mM.

Line 171-2: do the authors mean ‘or enhance alanine binding’?

Reviewer #2: The manuscript by Ishida et al. studied the range of chemicals recognized by T1r2a/T1r3, one of the taste receptor type I. Although the structure of the ligand binding domain of the T1r2a/T1r3 (T1r2a/T1r3LBD) has already been determined by the authors, the ligand recognition mechanism by the T1r2a/T1r3LBD has not yet been fully explored. The authors analyzed the binding of various chemicals to medaka T1r2a/T1r3LBDs using their developed assay system with purified medaka T1r2a/T1r3LBD. The authors validated the specificity of this protein for L-α-amino acids and the importance of the α-amino and carboxy groups in receptor recognition through a series of binding assays for amino acid derivatives. In addition to that, they discovered that the binding ability to the receptor is not limited to proteinogenic amino acids, but also to non-proteinogenic amino acids such as metabolic intermediates. In summary, I believe that this work would be an excellent contribution to PLOS ONE. My comments to improve the manuscript are below.

MAIN CONCERN

1. Although the authors have measured the binding affinities for many chemicals, they did not refer to their determined structure except L-Ala in Fig. 7. I believe that it would become a better paper if they could use the structure to discuss all chemicals. For example, can the all chemicals they measured fit into the ligand-binding pocket of the T1r2a/T1r3LBD or not? I suspect that IMP (Fig. 2A) does not fit into the pocket.

2. The authors performed differential scanning fluorimetry (DSF) to measure the binding affinity of the chemicals. Although they described that their assays were performed as described previously (ref 17), when I checked ref 17, there was no detailed description of the method (e.g. protein concentration, dye concentration, etc.). Since this manuscript is primarily concerned with measurements of the binding of chemicals with DSF, the methods should be accurately described in the manuscript.

3. In the case of IMP (Fig. 2C), citric acid (Fig. 4B), thiamine (Fig. 5B), serotonin (Fig. 6B), and ATP (Fig. 6B), ΔTm values were negative. However, there is no description or discussion of the effects of these chemicals on ΔTm. Especially, the effects of citric acid, serotonin, and ATP are so large that they should not be ignored and should be discussed.

4. The authors showed in Fig. 2A that the addition of IMP enhanced the binding affinity to L-Ala. Although this phenomenon has been described in the manuscript, there is no discussion as to why this effect occurred. Perhaps IMP has an allosteric effect on T1r2a/T1r3LBDs, but in any case, some discussions should be needed.

Minor concern

In Fig. 7, dotted lines are drawn as hydrogen bonds, but the length of each bond should be described to show that they are hydrogen bonds.

Reviewer #3: This is a continuation of a line of research that is focused on the structural and functional analysis of taste receptor in the Yamashita lab. In this manuscript, they further characterize the ligand specificity of the medaka T1r2a/T1r3 receptor. As expected, the receptor show specificity toward l-a-amino acids and the authors have demonstrated the importance of a-amino and carboxy groups for ligand recognition. Although a large portion of data shows negative results, such as they could not find any ligands of medaka T1r2/T1r3 common to the other T1rs and class C GPCRs other than L-amino acids, it still adds certain value to the literature. The manuscript is technically sound, and the manuscript is well written and easy to follow.

6. PLOS authors have the option to publish the peer review history of their article (what does this mean?). If published, this will include your full peer review and any attached files.

Reviewer #1: No

Reviewer #2: No

Reviewer #3: **Yes: **Peihua Jiang

---

## [Author Response · Author response to Decision Letter 0]

4 Mar 2024

Reviewer #1:

Ishida et al reports on binding assays aimed at determining the ligand binding specificity of the medaka fish T1r2a/T1r3 ligand binding domain (LBD). The crystallographic structure of this dimer has been solved, which enables more educated inferences from this study. The study is carefully conducted, and they use a variety of structurally diverse alanine derivatives, non-proteogenic amino acids, and structurally unrelated sugars, neurotransmitters, hormones, and vitamins. They conclude that only L amino acids are capable of binding to the LBD, and that substitutions to the carboxyl and amino groups are more tolerated compared to the substitution of the hydrogen at the alpha carbon. The paper is well written, and the conclusions are well supported by the experiments.

Reply: 

The authors thank the reviewer for their positive and useful comments on the manuscript.

My only comment is that they have used very low concentrations of glucose and sucrose. Although these are not natural ligands of fish T1rs, sucrose is known to generate some nerve response at a high concentration of around 300 mM.

Reply:

We thank the reviewer for their comment based on the taste physiology. Because the current methodology measures protein stabilization induced by ligand-binding, it is difficult to analyze the specific interactions of sugars at high concentrations, which intrinsically induces protein stabilization even without any specific interaction with the protein. We clarified this point in the section “Binding of ligands for the other T1rs or class C GPCRs” in the Results sections as follows:

(l.170–176, p.8)

“It should be noted that it is not feasible to assess the specific binding of sugars at high concentrations, such as those inducing responses of mammalian T1r2/T1r3 (~300 mM) [2, 4], using DSF, because they intrinsically induce thermal stabilization of a protein, even without specific interactions [22]. Nevertheless, the observation in this study, which showed no noticeable thermal stabilization by sugars up to at least 10 mM, is consistent with the results of a previous study that reported no responses of this receptor to sugars (at 150 mM) [5].”

Related to this change, we have added the following reference to the revised manuscript.

“22. Ajito S, Iwase H, Takata SI, Hirai M. Sugar-Mediated Stabilization of Protein against Chemical or Thermal Denaturation. J Phys Chem B. 2018;122:8685-97. doi: 10.1021/acs.jpcb.8b06572.”

Line 171-2: do the authors mean ‘or enhance alanine binding’?

Reply:

We thank the reviewer for pointing out that the original description was confusing. We intended to mean that IMP DID NOT enhance alanine binding, and revised the sentence as follows:

(l.178-179, p.8)

“However, in medaka T1r2a/T1r3LBD, IMP did not bind, and its presence did not enhance alanine binding (Fig 2B).”

Reviewer #2:

The manuscript by Ishida et al. studied the range of chemicals recognized by T1r2a/T1r3, one of the taste receptor type I. Although the structure of the ligand binding domain of the T1r2a/T1r3 (T1r2a/T1r3LBD) has already been determined by the authors, the ligand recognition mechanism by the T1r2a/T1r3LBD has not yet been fully explored. The authors analyzed the binding of various chemicals to medaka T1r2a/T1r3LBDs using their developed assay system with purified medaka T1r2a/T1r3LBD. The authors validated the specificity of this protein for L-α-amino acids and the importance of the α-amino and carboxy groups in receptor recognition through a series of binding assays for amino acid derivatives. In addition to that, they discovered that the binding ability to the receptor is not limited to proteinogenic amino acids, but also to non-proteinogenic amino acids such as metabolic intermediates. In summary, I believe that this work would be an excellent contribution to PLOS ONE. My comments to improve the manuscript are below.

Reply:

The authors thank the reviewer for their positive comments on the study. We appreciate the comments below, which were useful for revising the manuscript.

MAIN CONCERN

1. Although the authors have measured the binding affinities for many chemicals, they did not refer to their determined structure except L-Ala in Fig. 7. I believe that it would become a better paper if they could use the structure to discuss all chemicals. For example, can the all chemicals they measured fit into the ligand-binding pocket of the T1r2a/T1r3LBD or not? I suspect that IMP (Fig. 2A) does not fit into the pocket.

Reply:

The authors thank the reviewer for their valuable suggestion. We performed the docking simulations for the compounds of which we found the obvious binding to T1r2a/T1r3LBD in this study. We have added the results as S2 Fig and referred to them in the Discussion section of the main text as follows:

“S2 Fig. Docking simulations of the compounds revealed binding to T1r2a/3LBD in this study. (A) A close-up view of the ligand-binding site in T1r2a in the crystal structure of the L-alanine-bound form (PDB ID: 5X2N). (B–D) Representative docking poses of N-methyl-L-alanine (N-Me-L-Ala; ZINC 901468; B), L-alanine methyl ester (L-Ala Me ester; ZINC 34702232; C), and L-citrulline (ZINC 1532614; D) in the ligand-binding site of T1r2a. In panels B–D, the docking simulations were performed using SwissDock (Grosdidier et al. Nucleic Acids Res. 39, W270, 2011), using the coordinates of T1r2aLBD (PDB ID: 5X2N, chain A) without ligands as a target.”

(l.275–277, p.12)

“Docking simulations showed that these compounds could be accommodated in the ligand-binding site in a manner similar to the that of known ligands (e.g., L-alanine) without steric clashes (S2 Fig).”

For the reviewer’s information, we also performed a docking simulation for IMP and found that no docking poses were observed in the ligand-binding pocket, at least under the same conditions as those used in the simulation in S2 Fig, in which the protein model was treated as a rigid body.

Figure. Overlay of all docking poses of IMP (shown in yellow) on T1r2aLBD. The position of the ligand-binding pocket was indicated with a box in dashed lines.

Nevertheless, docking simulations often provide false docking poses for any chemical, even for those that do not show specific binding. Conversely, a simulation under current conditions does not provide a true docking pose if the binding of the ligand requires a conformational change from the input structure. Therefore, we refrained from showing the results of docking simulations for all chemicals and only showed that the compounds that showed binding in this study were capable of binding without a conformational change in the protein taking the conformation in complex with the known ligands.

2. The authors performed differential scanning fluorimetry (DSF) to measure the binding affinity of the chemicals. Although they described that their assays were performed as described previously (ref 17), when I checked ref 17, there was no detailed description of the method (e.g. protein concentration, dye concentration, etc.). Since this manuscript is primarily concerned with measurements of the binding of chemicals with DSF, the methods should be accurately described in the manuscript.

Reply: 

We thank the reviewer for pointing out the lack of information. We have added this information to the Materials and Methods section as follows:

(l.99–102, p.5)

“The sample protein was then mixed with Protein Thermal Shift Dye (Applied Biosystems) and the ligands in the dialysis buffer used for the final dialysis step. Specifically, 1 µg of protein, 1× Dye, and ligands at a concentration for each condition were mixed in 20 µL of reaction mix.” 

3. In the case of IMP (Fig. 2C), citric acid (Fig. 4B), thiamine (Fig. 5B), serotonin (Fig. 6B), and ATP (Fig. 6B), ΔTm values were negative. However, there is no description or discussion of the effects of these chemicals on ΔTm. Especially, the effects of citric acid, serotonin, and ATP are so large that they should not be ignored and should be discussed.

Reply:

We appreciate the reviewer’s important suggestion. We added the discussion about the relationship between the ΔTm values and the actions on the receptor in the Discussion section as follows:

(l.318–325, p.14)

“Although the addition of several compounds such as IMP, citric acid, thiamine, serotonin, and ATP provided negative ΔTm values, their actions on the protein are unlikely related to the binding that induces receptor responses, but probably destabilization of the protein due to their acidic/basic properties, because the correlation between the positive ΔTm values and the receptor responses has been reported [17]. On this account, a representative monoamine neurotransmitter serotonin did not show binding expected to induce receptor responses (Fig 6B), which is consistent with the results that T1r2a/T1r3 prefers α-amino acids and not monoamines (ethylamine; Fig 1).”

4. The authors showed in Fig. 2A that the addition of IMP enhanced the binding affinity to L-Ala. Although this phenomenon has been described in the manuscript, there is no discussion as to why this effect occurred. Perhaps IMP has an allosteric effect on T1r2a/T1r3LBDs, but in any case, some discussions should be needed.

Reply:

Thank you for pointing this out. We realized that the original description caused confusion. We meant to state that IMP DID NOT enhance alanine binding. To avoid confusion, we revised the sentence as follows:

(l.178-179, p.8)

“However, in medaka T1r2a/T1r3LBD, IMP did not bind, and its presence did not enhance alanine binding (Fig 2B).”

Minor concern

In Fig. 7, dotted lines are drawn as hydrogen bonds, but the length of each bond should be described to show that they are hydrogen bonds.

Reply:

We appreciate the reviewer’s valuable suggestion. We have revised the Fig. 7A and 7B in accordance with their suggestion. Accordingly, the figure legends have been revised as follows:

(l.282–283, p.12)

“The dashed lines depict the hydrogen bonds labeled with their distances in Å.”

Reviewer #3:

This is a continuation of a line of research that is focused on the structural and functional analysis of taste receptor in the Yamashita lab. In this manuscript, they further characterize the ligand specificity of the medaka T1r2a/T1r3 receptor. As expected, the receptor show specificity toward l-a-amino acids and the authors have demonstrated the importance of a-amino and carboxy groups for ligand recognition. Although a large portion of data shows negative results, such as they could not find any ligands of medaka T1r2/T1r3 common to the other T1rs and class C GPCRs other than L-amino acids, it still adds certain value to the literature. The manuscript is technically sound, and the manuscript is well written and easy to follow.

Reply:

The authors thank the reviewer for the positive evaluation of the study.

---

## [Decision Letter · Decision Letter 1]

8 Mar 2024

Chemical range recognized by the ligand-binding domain in a representative amino acid-sensing taste receptor, T1r2a/T1r3, from medaka fish

PONE-D-23-32338R1

Dear Dr. Yamashita,

We’re pleased to inform you that your manuscript has been judged scientifically suitable for publication and will be formally accepted for publication once it meets all outstanding technical requirements.

Kind regards,

Israel Silman

Academic Editor

PLOS ONE

Additional Editor Comments (optional):

Reviewers' comments:

Reviewer's Responses to Questions

**Comments to the Author**

1. If the authors have adequately addressed your comments raised in a previous round of review and you feel that this manuscript is now acceptable for publication, you may indicate that here to bypass the “Comments to the Author” section, enter your conflict of interest statement in the “Confidential to Editor” section, and submit your "Accept" recommendation.

Reviewer #2: All comments have been addressed

2. Is the manuscript technically sound, and do the data support the conclusions?

Reviewer #2: Yes

3. Has the statistical analysis been performed appropriately and rigorously? 

Reviewer #2: Yes

4. Have the authors made all data underlying the findings in their manuscript fully available?

Reviewer #2: Yes

5. Is the manuscript presented in an intelligible fashion and written in standard English?

Reviewer #2: Yes

6. Review Comments to the Author

Reviewer #2: The author responded very well to my four main concerns and one minor one, so I have no other specific comments on the revised manuscript.

7. PLOS authors have the option to publish the peer review history of their article (what does this mean?). If published, this will include your full peer review and any attached files.

Reviewer #2: **Yes: **Haruo Ogawa

---

## [Editor Report · Acceptance letter]

13 Mar 2024

PONE-D-23-32338R1 

PLOS ONE

Dear Dr. Yamashita, 

I'm pleased to inform you that your manuscript has been deemed suitable for publication in PLOS ONE. Congratulations! Your manuscript is now being handed over to our production team.

Kind regards, 

on behalf of

Prof. Israel Silman 

Academic Editor

PLOS ONE